# Could Fibrin Sealants (TISSEEL^TM^) Be Effective in the Management of Burn Injuries? A Histopathological Study in Rats

**DOI:** 10.3390/medsci12040075

**Published:** 2024-12-19

**Authors:** Christina Nikolaou, Maximos Frountzas, Dimitrios Schizas, Vasilios Pergialiotis, Emmanouil I. Kapetanakis, Konstantinos Kontzoglou, Despina N. Perrea, Efthymios Koniaris, Stylianos Kykalos, Dimitrios Iliopoulos

**Affiliations:** 1Laboratory of Experimental Surgery and Surgical Research “N.S. Christeas”, School of Medicine, National and Kapodistrian University of Athens, 11527 Athens, Greece; kristenni@med.uoa.gr (C.N.); pervasilis@med.uoa.gr (V.P.); kckont@med.uoa.gr (K.K.); dperrea@med.uoa.gr (D.N.P.); diliop@med.uoa.gr (D.I.); 2Department of Plastic & Reconstructive Surgery, Burn Center, General Hospital of Athens “G. Genimatas”, 11527 Athens, Greece; 3First Department of Surgery, “Laikon” General Hospital, School of Medicine, National and Kapodistrian University of Athens, 11527 Athens, Greece; schizasad@gmail.com; 4First Department of Obstetrics and Gynecology, Unit of Gynecologic Oncology, “Alexandra” Hospital, School of Medicine, National and Kapodistrian University of Athens, 11527 Athens, Greece; 5Department of Thoracic Surgery, “Attikon” University Hospital, School of Medicine, National and Kapodistrian University of Athens, 12461 Athens, Greece; 6Second Department of Propaedeutic Surgery, “Laikon” General Hospital, School of Medicine, National and Kapodistrian University of Athens, 11527 Athens, Greece; skykalos@med.uoa.gr; 7First Department of Pathology, “Hippocration” General Hospital, 11527 Athens, Greece; e.koniaris@hippocratio.gr

**Keywords:** TISSEEL, rat animal model, burn, treatment, experimental

## Abstract

**Background:** Burn injuries remain a major clinical problem worldwide, which require special management by experienced plastic surgeons. However, they cannot be available in every healthcare unit; consequently, there is a need for effective treatment options that could be utilized by a wide range of non-expert healthcare professionals. The aim of the present experimental study was to investigate the safety and efficacy of using a fibrin sealant (TISSEEL^TM^) compared to the conventional treatment with sulfadiazine on partial-thickness burn in a rat animal model. **Methods:** A cohort of Sprague Dawley rats underwent partial-thickness contact thermal burn wounds and were divided into three study groups: control group (no treatment), silver sulfadiazine cream group and TISSEEL^TM^ group. Following animal sacrifice, a blinded histopathologic analysis was conducted regarding inflammatory response, healing and tissue regeneration. **Results:** In total, 30 animals were included with a median weight of 236 ± 10 g. Two animals from the control group died on the first postoperative day. Animals in the TISSEEL^TM^ group presented dominant collagen expression compared to animals in the control and silver sulfadiazine cream group (*p* = 0.000). Histopathologic analysis also demonstrated marked leukocyte infiltration (*p* = 0.009), increased neovascularization (*p* = 0.000) and higher fibroblast expression (*p* = 0.002) in the TISSEEL^TM^ group compared to the other two groups. **Conclusions:** TISSEEL^TM^ seems to be a safe alternative (or even principal) option for the initial therapeutic approach of partial-thickness burn injuries. Moreover, it seems to be superior to silver sulfadiazine in terms of tissue healing and regeneration. However, additional experimental as well as clinical research is necessary prior to implementation in clinical practice.

## 1. Introduction

Burn injuries remain a clinical challenge for all healthcare systems worldwide consuming great amounts of resources both in the emergency and the rehabilitation setting due to demands of wound healing and reconstruction and the eventual return to daily activity of patients [1]. Costs of providing direct care for burns tend to be quite high, reaching hundreds of thousands of dollars in long lasting hospitalizations [2,3]. In addition, burn injuries usually lead to increased length of hospital stay (LOS), which is proportional to the total body surface area (TBSA) of burn injury, the presence of inhalation injury, the patients’ age and the healthcare-associated infections (HAIs) [4,5]. Finally, the mortality rate for most hospitalized patients with severe burn injuries typically ranges from 1.4% to 18% (with a maximum incidence of 34%), whereas in Europe there are 180,000 deaths due to severe burn injuries per year [6]. The risk of death is increased 7-fold in low- and middle-income countries compared to high-income ones [7].

The most crucial step for burn treatment and prognosis is a reliable assessment of the extent and depth of the burn injury by surveying the calibrated areas of the skin and recording any changes. Alternatively, another simple, quick, safe and reproducible clinical procedure for assessing the burn area is to check perfusion by needle bleeding. Other techniques of classifying burns include Laser Doppler methods (such as Laser Doppler Imaging and Laser Doppler Perfusion Imaging), radiography, fluoroscopy, high-frequency ultrasonography (hypertrophic B mode), magnetic resonance imaging and choroidal chorionic villus sampling [8]. Burn injuries are categorized as superficial (first-degree), which involve only the epidermis, and deep, which include partial-thickness (second-degree), that affect the superficial layer of the dermis, and full-thickness (third-degree), which are the most severe and affect both the epidermis and dermis skin layers, but can also extend into the subcutaneous tissue.

The gold standard treatment for deep acute burn injuries involves a series of interventions tailored to each burn degree and any associated complications or comorbidities. Partial-thickness deep burn injuries necessitate the immediate removal of burned tissue and coverage of the burn surface with sterile gauzes or foam dressings to reduce the risk of exogenous infection, protein and fluid loss, as well as life-threatening hypothermia [9]. Treatment also aims to prevent adjacent tissue damage by using antimicrobial creams and occlusive dressings to create a moist environment that promotes faster healing. So far, silver sulfadiazine cream has been the most widely applied agent for partial-thickness burns, demonstrating antimicrobial and regenerative properties with great efficacy by changing dressing every other day [10]. Nevertheless, specialized plastic surgeons capable in handling such complex conditions cannot be available in the emergency setting of every healthcare unit, especially in developing countries [11]. In addition, general surgeons usually have limited or no expertise in burn therapy. Under these circumstances, there exists the need for a widely available, easily applied and effective treatment option for burn injuries.

Fibrin sealants have shown promising hemostatic and regenerative properties, such as reduced hematoma formation, lower graft dislocation rates and higher graft take rates, in experimental graft fixation models [12]. The aim of the present experimental study was to investigate the safety and effectiveness of the fibrin sealant TISSEEL^TM^ as an alternative treatment approach for immediate partial-thickness burn injuries, in terms of histopathological assessment of the local inflammatory reaction and associated regenerative process, as well as evaluating the subject animals’ clinical course. Potentially, this could allow any healthcare professional, regardless of experience level, to properly handle a burn injury and serious sequalae could be avoided.

## 2. Materials and Methods

### 2.1. Ethical Approval

The experimental protocol was firstly approved by the Athens University Medical School’s Ethics Committee and subsequently it obtained veterinary animal use approval by the Veterinary Service of the Prefecture of Attica, Greece (Ref. No 345/29.09.2020). Animal care, surgical procedures and postoperative recovery complied with the Animal Research Reporting of In Vivo Experiments (ARRIVE) guidelines and the European Directive 2010/63 on the protection of animals used for scientific purposes. The researchers adhere to the Three Rs (replacement, reduction and refinement) principle in laboratory animal experiments proposed by Russel and Birch in 1959.

### 2.2. Animal Care and Conditions

All experimental procedures were conducted in the Laboratory of Experimental Surgery and Surgical Research “N.S. Christeas” at the Medical School of the National and Kapodistrian University of Athens. Standardized Sprague Dawley rats procured from the National Centre for Scientific Research “Demokritos” were used in the experiment. The animals had been acclimated to their new surroundings in atmosphere-controlled chambers with controlled light levels (12 h day/night cycle), temperature range (20 ± 1 °C) and humidity (55 ± 5%) for seven days prior to the experiment. Full nutritional supplementation with ELVIZ 510 food pellets was utilized.

### 2.3. Fibrin Sealant

The biocompatible agent TISSEEL^TM^ (Baxter Healthcare Corporation, Deerfield, IL, USA) comprises two sterile, deep-frozen solutions containing a combination of coagulation factors: the thrombin solution and the sealer protein solution. Each solution is contained in a preloaded, separate double-chamber syringe. The thrombin solution comprises human thrombin and calcium chloride as active components, fractionated from pooled human plasma, whereas the sealer protein solution contains synthetic aprotinin, factor XIII and fibrinogen [7]. In order to create a stable thrombus at the appliance site, TISSEEL^TM^ mimics the last stages of the coagulation cascade. Specifically, fibrogen is broken down by thrombin into fibrin monomers, which along with coagulation factor XIII form a fibrin thrombus that is stabilized by the action of aprotinin, which prevents its breakdown [13]. Before using the agent, the thrombin solution and the sealer protein solution were gently heated back up to 37 °C in a water bath and mixed in a common syringe immediately prior to application.

### 2.4. Experimental Protocol and Surgical Procedure

Intraperitoneal administration of Ketamine (60 mg/kg) and Xylazine Hydrochloride (5 mg/kg) was performed so as to anesthetize the animals prior to the procedure. Xylazine is an agonist at the α-2 adrenergic receptors and a clonidine analogue. Ketamine is used to create and sustain anesthesia, by producing sedation, analgesia and memory loss [14]. As soon as the rats were fully anesthetized, their dorsum’s hair was shaved off using an electrical shaver. The dorsum was selected as the burn injury area, because it is inaccessible by the animal and therefore further trauma induced by biting or scratching would be avoided. A square metal plate measuring 2.5 cm long by 2.5 cm wide and 6 mm deep with a fireproof handle was used to create a partial-thickness burn injury at the dorsal region of each animal. Using a special thermocoupler, the metal plate was heated in a burner up to 100 °C and then placed onto the animal’s shaved area with minimal pressure applied for 10 s [15,16]. This procedure was followed for all experimental animals, resulting in a partial-thickness direct contact burn.

Animals were randomly divided into three groups of ten (Figure 1): Animals in group 1 (control group) received no treatment on the burn surface. In group 2 (sulfadiazine group), 0.5 mL silver sulfadiazine cream was used to cover the burn surface. In group 3 (TISSEEL^TM^ group), 0.5 mL of a biological adhesive-sealing fibrin (TISSEEL^TM^) was applied on the burn surface. Immediately after the procedure, all experimental animals received regularly scheduled subcutaneous Tramadol (0.01 mg/kg) for pain control. All experimental animals were euthanized under ether anesthesia on the 10th day post-burn. The 10-day experiment interval provided an adequate time period to investigate the feasibility of utilizing a novel treatment option (TISSEEL^TM^) and compare it to the conventional therapeutic choice (silver sulfadiazine) in terms of the initial histopathologic consequences after its application, which was the aim of the present experimental study. Subsequently, a square (2 cm each side) section of skin and underlying tissues at the burn surface orifice of each animal was excised and prepared for histological analysis. Surgical specimens were then fixed in 10% formaldehyde solution and histopathological analysis followed.

### 2.5. Pilot Experiment and Procedure Standardization

Before the main experiment, a pilot study involving four animals was carried out to standardize the model of burn injury and assess any animal reactions to TISSEEL^TM^ application after burn injury. The first rat suffered an untreated contact thermal burn injury at 100 °C for 10 s resulting in deep thickness burn, which was left untreated and they died on the 2nd postoperative day. The second rat underwent a contact thermal burn at the same temperature, but for 5 s, resulting in a partial-thickness burn, which was also left untreated. This animal was euthanized on the 5th postoperative day under ether anesthesia. The same model of burn injury was applied in the other two animals, which were treated with TISSEEL^TM^ application after partial-thickness burn injuries. No post-procedure reactions were observed and the animals were euthanized on the 5th postoperative day under ether anesthesia as well. The experimental design was standardized based on previous animal burn wound methods in the literature [15,16].

### 2.6. Histopathological Analysis

The histopathological analysis examining inflammatory infiltration, tissue healing and regeneration was performed by two experienced, consultant histopathologists who blindly reviewed and reported on all fixed specimens. After fixation in 10% formaldehyde solution, surgical specimens were embedded in paraffin blocks. Three non-consecutive sections per animal and 5 fields per section were analyzed. Due to the lack of a standardized histopathological score concerning inflammatory infiltration, tissue healing and regeneration of burn wounds in experimental models, an adaptation of existing scores related to the investigated parameters was attempted by the two histopathologists. Under these circumstances, an analytical evaluation of each section according to specific histopathologic features was not possible. Thus, each animal was included in the histopathological category with the most common criteria from the exiting histopathological scores, which will be analyzed later (Table 1, Table 2, Table 3 and Table 4).

Hematoxylin–eosin (H&E) staining was utilized, as dermal structures, such as glandular structures and hair follicles, are readily distinguishable using H&E. Inflammatory cells are also quite visible, despite being smaller in size, allowing easy distinction from other dermal cells. A modified histologic scoring system of leucocytes’ presence was created based on a previously reported scoring system [17,18,19]. According to that scoring system, mild (<30%) leucocyte infiltration was scored with 1 point, moderate leucocyte infiltration (30–60%) was scored with 2 points and marked leukocyte infiltration was scored with 3 points (Table 1).

Additionally, fibroblastic proliferation was identified using Vimentin staining, which is the best marker for fibroblast identification [20,21]. Scoring was based on the number of fibroblasts in the wound: fibroblasts’ proliferation in 0–40% of the wound area was scored with 1 point, fibroblasts’ proliferation in 40–80% of the wound area was scored with 2 points and fibroblasts’ proliferation in >80% of the wound area was scored with 3 points (Table 2).

Another important histopathological parameter was neovascularization, which was observed using CD-31, which is an important marker for endothelial cells [22]. To assess neovascularization, the scoring system proposed by Abramov et al. was used [23]. According to that scoring system, the absence of newly formed vessels was scored with 1 point, the presence of 1–5 vessels was scored with 2 points, the presence of 6–10 vessels was scored with 3 points and the presence of >10 vessels was scored with 4 points (Table 3).

Lastly, collagen deposition was also measured using H&Ε staining, because the quality of staining using the Masson’s trichrome stain, which seems to be the most effective in collagen staining, was very low due to the damage in burn tissue [24]. Focal collagen deposition around capillaries was scored with 1 point, moderate collagen deposition around repair tissue was scored with 2 points and dominant collagen deposition was scored with 3 points (Table 4).

### 2.7. Statistical Analysis

SPSS version 25.0 software for Macintosh was used for the statistical analyses. Continuous variables were presented as mean ± standard deviation. The Kolmogorov–Smirnov test and graphical techniques were used to evaluate the normality of distributions. The non-parametric Mann–Whitney test was utilized to compare means between groups and the chi-square, as well as Fisher’s exact tests were used to compare proportions. Using chi-square and Fisher’s exact tests, post hoc analysis was carried out on the quantitative variables of the histopathological analysis. The statistical significance threshold is set at *p* < 0.05, and all reported *p*-values are two-tailed. A post hoc power analysis was conducted to assess the sample size power [25].

## 3. Results

### 3.1. Animal Characteristics

In total, 30 Sprague Dawley rats were included in this study. Their mean weight was 236 ± 10 g and their age ranged between 21 to 25 weeks old. There were no differences regarding the animals’ weight or age at the time of randomization. Burn wound surface was covered with silver sulfadiazine cream in 10 animals (sulfadiazine group), in 10 animals (TISSEEL^TM^ group) a biological adhesive-sealing fibrin (TISSEEL^TM^) was applied and finally, 10 animals (control group) received no treatment (Figure 1).

### 3.2. Postoperative Outcomes

Apart from two animals of the control group, which passed away 12 h after the experiment, the postoperative course of the other animals was uneventful. There were neither signs of TISSEEL^TM^ rejection reactions nor postoperative complications observed; therefore, all animals were euthanized on the 10th POD as per protocol. In addition, during macroscopic evaluation of the experimental animals after the application of therapeutic options subsequent to the partial-thickness burn injury, a stable thrombus formation was observed in the TISSEEL^TM^ group, which led in diminishing the burn-associated hemorrhage (Figure 2).

### 3.3. Histopathological Assessment

During histopathological analysis, hematoxylin and eosin (H&E) staining was utilized for the assessment of inflammatory infiltration based on the number of leucocytes per high-power field. Marked leucocyte infiltration was observed in 6 out of 10 animals (60%) of the TISSEEL^TM^ group compared to 2 out of 10 animals (20%) of the sulfadiazine group and 1 out of 8 animals (12.5%) of the control group (*p* = 0.009, Table 5, Figure 3). A statistically significant difference was also observed during the inter-group comparison of TISSEEL^TM^ with the conventional treatment options (60% vs. 17%, *p* = 0.012, Table 5).

In addition, fibroblastic proliferation was also more prominent in the TISSEEL^TM^ group, where in 5 out of 10 animals (50%) fibroblasts were present in >80% of the wound area, whereas in the sulfadiazine group, only 3 out of 10 animals (30%) demonstrated fibroblasts’ growth in >80% of the wound area and in the control group, none of the 8 animals (0%) had fibroblasts in >80% of the wound area (*p* = 0.002, Table 5, Figure 4). However, a statistically significant difference was not observed during the inter-group comparison of TISSEEL^TM^ with the conventional treatment options (50% vs. 17%, *p* = 0.087, Table 5).

The TISSEEL^TM^ group presented a higher percentage of newly formed vessels, with 9 out of 10 (90%) animals, demonstrating more than 10 newly formed vessels, whereas in the sulfadiazine group, only in 1 out of 10 (10%) animals, more than 10 newly formed vessels were observed and in the control group, none of the 8 animals (0%) presented more than 10 newly formed vessels (*p* = 0.000, Table 5, Figure 5). A statistically significant difference was also observed during the inter-group comparison of TISSEEL^TM^ with the conventional treatment options (90% vs. 6%, *p* = 0.012, Table 5).

Finally, collagen deposition was higher in the TISSEEL^TM^ group, in which 7 out of 10 (70%) animals presented with dominant collagen concentration, while in the sulfadiazine group, only 1 out of 10 (10%) animals presented dominant collagen concentration, while in the control group none of the 8 animals (0%) presented dominant collagen concentration (*p* = 0.000, Table 5, Figure 6). A statistically significant difference was also observed during the inter-group comparison of TISSEEL^TM^ with the conventional treatment options (70% vs. 6%, *p* = 0.001, Table 5).

## 4. Discussion

Burn injuries represent a complex medical problem in all healthcare systems and can lead to direct tissue damage and extensive hemorrhage, as well as indirect consequences, such as secondary infections due to external skin barrier destruction. Therefore, the modern therapeutic approach with skin implants and tissue flaps seems to offer the highest success rates, but it frequently necessitates the expertise of specialized medical professionals/plastic surgeons.

Nevertheless, most healthcare units typically do not provide specialized burn injury services, leading to high morbidity and mortality rates. Under these circumstances, there is a need for a safe and effective immediate treatment option for acute burn injuries, which could serve as a “damage control” strategy, until definite specialized treatment is provided. In this experimental animal model study, TISSEEL^TM^ seemed to be a safe option for partial-thickness burn injury management, with no adverse effects observed after its use. Histopathologic analysis revealed a clear superiority of TISSEEL^TM^ compared to sulfadiazine, which has been the current, conventional treatment option for partial-thickness burn injuries, in regards to tissue healing. Similarly, TISSEEL^TM^ application demonstrated superior histopathological responses compared to no treatment at all. In particular, TISSEEL^TM^ was associated with more prominent leukocyte infiltration at the burn area indicating a more intense inflammatory process, which is the first step prior to tissue proliferation, which leads to wound healing [26]. Moreover, TISSEEL^TM^ application led to higher fibroblastic proliferation rates at the burn area showing greater healing activity. Finally, TISSEEL^TM^ was superior to sulfadiazine regarding tissue regenerative process as well, as it was related to elevated rates of newly formed vessels and greater collagen deposits at the site of burn injury.

Fibrin sealants have already shown hemostatic and adhesive properties in several experimental surgical models. Current evidence suggests that fibrin glue has a hemostatic effect and neovascularization properties compared to epinephrine tumescent, which was less effective in preventing re-bleeding after endoscopic therapy [27]. Moreover, experimental studies on colon perforations have demonstrated that fibrin glues have sealing properties by promoting collagen deposition, while exhibiting reduced inflammatory and fibrotic reactions on histopathological analysis [28]. Finally, TISSEEL^TM^ seemed to be superior than conventional suturing in terms of inflammatory reaction, tissue healing and regenerative process in an experimental model of bowel perforation, while it showed lower hemorrhagic infiltration compared to sutures [29].

Apart from hemostatic and healing properties, fibrin sealants could be helpful in graft fixation, as it has been demonstrated in several burn animal models. Initially, fibrin sealants seem to promote graft adhesion in experimental studies compared to conventional suture fixation (99.7% vs. 95.9%), while they are also associated with lower postoperative complications, such as hematoma and seroma [30]. Likewise, Vedung et al. demonstrated decreased inflammatory reaction and improved healing when fibrin sealants were used with skin grafting on contaminated burn wounds [31]. Similarly, Nervi et al., when he evaluated the efficacy of fibrin glue as a topical hemostatic agent, concluded that it significantly decreases the time to hemostasis at the donor skin harvest site in patients undergoing skin grafting [32]. Finally, Loor et al. showed that fibrin glue enhanced neovascularization during healing of wounds treated with dermal substitutes [33].

The anti-inflammatory properties of fibrin glues and fibrin sealants, as demonstrated in the aforementioned experimental models and clinical studies, seem to be contradictory to the present study’s findings of increased leukocyte infiltration after TISSEEL^TM^ application compared to silver sulfadiazine. However, the post-burn immune response is unique and could not be compared to anastomotic healing or skin graft regeneration processes [34]. The burn wound healing is largely affected by an adequate local inflammatory response to stop progression from stasis zone to coagulation zone at the burn area, which is associated with severe systemic complications, such as sepsis, hemodynamic failure and multiple organ dysfunction [35]. Under these circumstances, elevated inflammatory reaction at the burn area is considered as a beneficial characteristic of TISSEEL^TM^ in the initial phase of burn healing, which was investigated in our study. However, this would be investigated by future studies, which would assess the long-term histopathologic and clinical features considering TISSEEL^TM^ utilization for burn management.

To the best of our knowledge, this is the first experimental study that investigates the role of TISSEEL^TM^ as an immediate treatment option for partial-thickness burn injuries in a rat model. Not only does this study evaluate the clinical course, the macroscopic appearance and histopathological outcome after TISSEEL^TM^ application, but it compares these parameters with silver sulfadiazine, which is the conventional treatment option for partial-thickness burns. Another strength of the present study is the randomization process of experimental animals before experimental teams’ recruitment, as shown in Figure 1. Moreover, all experimental procedures were performed by a single operator and the histopathological evaluation was conducted by two blinded histopathologists, who were unaware of the burn treatment that each animal had received. Finally, the histopathological evaluation was based on previously established grading systems but adaptations were made according to rat histology as necessary to accurately assess inflammation, neovascularization, collagen deposition and fibrosis in the fixed specimens.

Despite being performed so as to adhere to the Three Rs principle, the reduced number of included animals investigated may be considered by some as a limitation of the study. However, the present work still maintains its scientific impact as it could serve as a preliminary investigation for further experimental or even clinical trials. Under these circumstances, a post hoc power analysis was performed and demonstrated a power of 65%, which is acceptable for a novel experimental study that investigates the safety and effectiveness of a burn treatment option for the first time [25].

Another potential limitation that could be considered is that the long-term clinical result of TISSEEL^TM^ application was not evaluated due to the sacrifice of the animals on the 10th postoperative day. Nonetheless, 10 days proved to be a sufficient time interval for the assessment of the initial histopathological response after TISSEEL^TM^ application, as well as the comparison with conventional treatment modalities. This was the principal aim of the present study. Long-term aesthetic outcomes and combination with definite treatment options, which require specialized burn injury service could be answered by future experimental and clinical trials.

Ultimately, TISSEEL^TM^ has been associated with an increased cost compared to the traditional management methods. Nevertheless, when evaluating the long-term benefit that an ideal initial management could offer in terms of inflammatory reaction, tissue healing and regeneration, one must consider that total medical costs would be diminished by preventing long hospital stays or the need to transfer to burn units by reducing the risk of burn wound complications like bleeding and infection.

Therefore, TISSEEL^TM^ seems to offer a safe and effective option for the initial management of a partial-thickness burn injury, showing better macroscopic outcomes in terms of hemorrhage and superior histopathological outcomes in terms of inflammatory infiltration, tissue healing and regeneration compared to the conventional local treatment options of such burns. Under these circumstances, it could offer an alternative, more efficacious treatment to “frontline” healthcare providers who are called upon to manage partial-thickness burns, prior to definite therapy by specialized plastic surgeons.

Nonetheless, the promising outcomes that have been demonstrated by the present study have to be confirmed by future experimental studies with a larger number of included animals, which would also investigate long-term local outcomes after TISSEEL^TM^ application both in terms of local histopathologic assessment and of systemic reaction (as evaluated by laboratory studies). In addition, clinical trials could be designed to investigate the role of TISSEEL^TM^ in the therapeutic algorithm of partial-thickness burn injury management and its combination with specialized surgical procedures such as early burn exposure and coverage with allografts or xenografts. Consequently, the current experimental study could be the basis for the design of future research and potential therapeutic applications of TISSEEL^TM^ on partial-thickness burn injuries.

## 5. Conclusions

In conclusion, TISSEEL^TM^ seemed to be a safe and effective alternative to the conventional treatment option of silver sulfadiazine application for the immediate treatment of burn injuries. Macroscopic hemorrhage was diminished after the burn injury using TISSEEL^TM^. Additionally, TISSEEL^TM^ proved to increase the inflammatory reaction, which is the initial step of tissue healing, after partial-thickness burns. Moreover, animals treated with TISSEEL^TM^ demonstrated increased fibroblast proliferation, neovascularization and collagen concentration compared to silver sulfadiazine cream. Therefore, TISSEEL^TM^ application could be implemented in clinical practice for the initial management of partial-thickness burn injuries. However, systematic reaction and long-term outcomes associated with its utilization should be investigated by larger scale, future experimental and clinical trials.

## Figures and Tables

**Figure 1 medsci-12-00075-f001:**
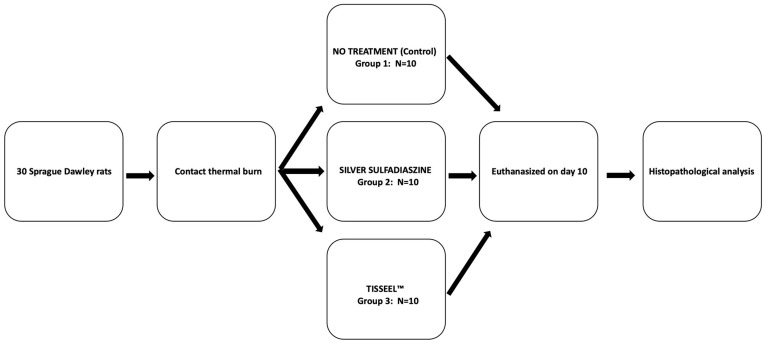
Flow diagram of the experimental design.

**Figure 2 medsci-12-00075-f002:**
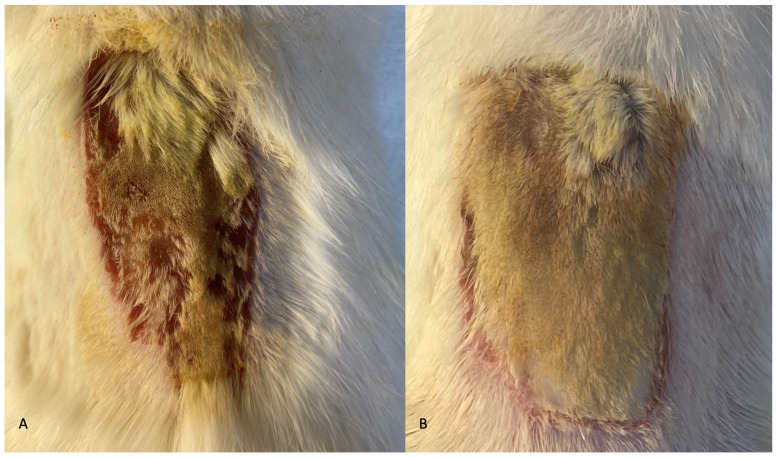
(**A**). Macroscopic evaluation of the burn surface after silver sulfadiazine application with marked burn-associated hemorrhage. (**B**). Macroscopic evaluation of the burn surface after TISSEEL^TM^ application without burn-associated hemorrhage.

**Figure 3 medsci-12-00075-f003:**
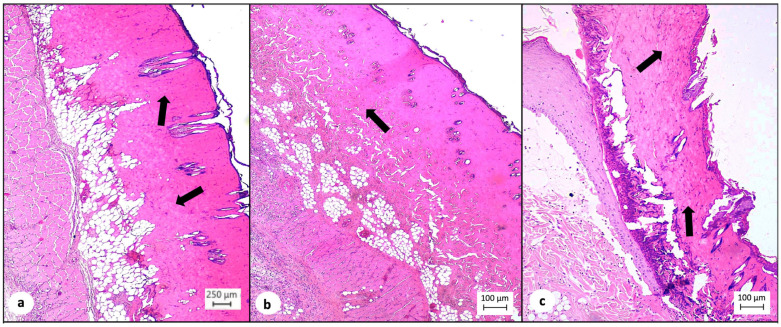
(**a**) Decreased number of lymphocytes (arrows) in the dermis of the control test animals (hematoxylin–eosin, ×10). (**b**) Moderate number of lymphocytes (arrow) in the dermis of test animals treated with silver sulfadiazine (hematoxylin–eosin, ×4). (**c**) Increased number of lymphocytes (arrows) in the dermis of test animals treated with TISSEEL^TM^ (hematoxylin–eosin, ×4).

**Figure 4 medsci-12-00075-f004:**
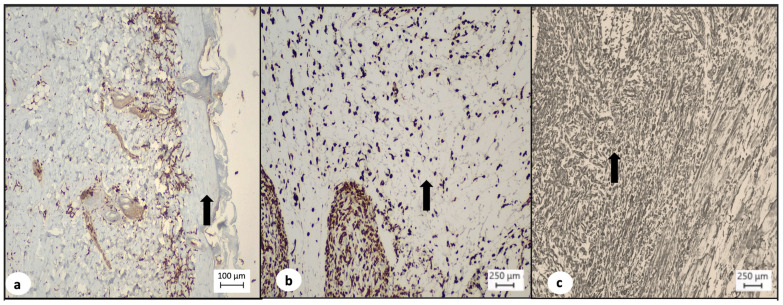
(**a**) Diminished fibroblast proliferation (arrow) in the dermis of control test animals (Vimentin, ×4). (**b**) Moderate fibroblast proliferation (arrow) in the dermis of test animals treated with silver sulfadiazine (Vimentin, ×10). (**c**) Marked fibroblast proliferation (arrow) in the dermis of test animals treated with TISSEEL^TM^ (Vimentin, ×10).

**Figure 5 medsci-12-00075-f005:**
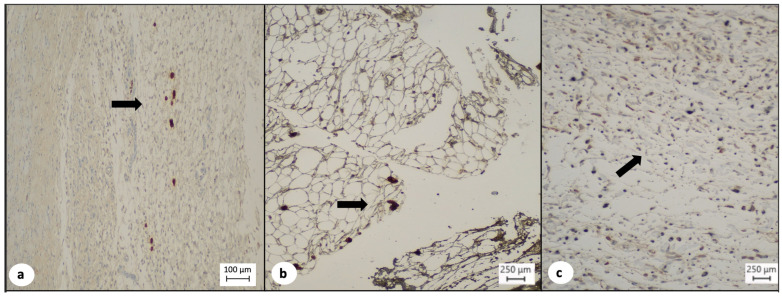
(**a**) Low expression of CD-31 (arrow) in the dermis of control test animals indicating decreased neovascularization (CD-31 immunochemistry, ×4). (**b**) Moderate expression of CD-31 (arrow) in the dermis of test animals treated with silver sulfadiazine indicating increased neovascularization (CD-31 immunochemistry, ×10). (**c**) Marked expression of CD-31 (arrow) in the dermis of test animals treated with TISSEEL^TM^ indicating marked neovascularization (CD-31 immunochemistry, ×10).

**Figure 6 medsci-12-00075-f006:**
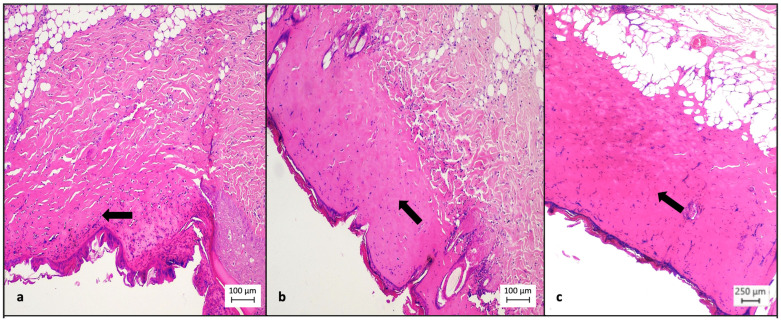
(**a**) Diminished collagen concentration in the dermis (arrow) of control test animals (hematoxylin–eosin, ×4). (**b**) Increased collagen concentration (arrow) in the dermis of test animals treated with silver sulfadiazine, but not arrayed in parallel (hematoxylin–eosin, ×4). (**c**) Increased collagen concentration (arrow) in the dermis of test animals treated with TISSEEL^TM^, which is in parallel, staggered arrangement, indicating more effective tissue healing (hematoxylin–eosin, ×10).

**Table 1 medsci-12-00075-t001:** A modified histologic grading scale in terms of tissue inflammation based on leucocyte concentration.

Inflammation Grading		
Leucocyte Infiltration	Mild: <30%	1
	Moderate: 30–60%	2
	Marked: >60%; dense infiltration	3

**Table 2 medsci-12-00075-t002:** A modified histologic grading scale in terms of fibroblastic proliferation using Vimentin staining.

Fibrosis Assessment		
Fibroblastic Proliferation	0–40% of wound	1
	40–80% of wound	2
	>80% of wound	3

**Table 3 medsci-12-00075-t003:** Histologic grading scale in terms of neovascularization depending on the presence of newly formed vessels.

Grading Scale for Neovascularization		
Neovascularization	None	1
	1–5 vessels	2
	6–10 vessels	3
	>10 vessels	4

**Table 4 medsci-12-00075-t004:** Histologic grading scale in terms of collagen deposition.

Tissue Healing		
Matrix deposition	Focal around capillaries	1
	Moderate in repair tissue	2
	Dominant	3

**Table 5 medsci-12-00075-t005:** Histopathological analysis of specimens regarding inflammatory infiltration, tissue healing and regeneration. *p*-values in bold represent statistical significance.

Histologic Parameters	Group 1 CONTROL	Group 2 SYLFIO	Group 3 TISSEEL^TM^	*p*-Value	
				1 vs. 2 vs. 3	3 vs. 1 & 2
Inflammation (Leukocyte infiltration)					
Mild (≤30%)	6	3	0	***p* = 0.009**	***p* = 0.012**
Moderate (30–60%)	1	5	4		
Marked (≥60%)	1	2	6		
Fibrosis (Fibroblastic proliferation)					
0–40% of wound	7	1	1	***p* = 0.002**	*p* = 0.087
40–80% of wound	1	6	4		
>80% of wound	0	3	5		
Neovascularization (CD31 expression)					
1–5 vessels	7	5	1	***p* = 0.000**	***p* = 0.000**
6–10 vessels	1	4	0		
>10 vessels	0	1	9		
Tissue healing (collagen deposition)					
Focal around capillaries	5	1	0	***p* = 0.000**	***p* = 0.001**
Moderate in repair tissue	3	8	3		
Dominant	0	1	7		

## Data Availability

All data are available upon reasonable request.

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
