# Peer review of "Could Fibrin Sealants (TISSEELTM) Be Effective in the Management of Burn Injuries? A Histopathological Study in Rats"

_medsci, 2024, doi:10.3390/medsci12040075_

Round 1
Reviewer 1 Report
Comments and Suggestions for Authors
1. Please shorten the initial part of the introduction concerning comparisons of burn treatment costs.
2. In the introduction, please describe in more detail the properties and role of fibrin sealants available on the medical market, to which the authors refer, citing, among others, article no. 12.
3. Is the only criterion for using silver sulfadiazine cream as an experiment control its widespread availability?
4. Are fibrin glues already used to treat burns in humans and if so, why was one of them not used in the experiment as a control?
Author Response
Please see attached response to reviewer 1 word file.

Reviewer 2 Report
Comments and Suggestions for Authors
The current manuscript assesses the efficacy of TISSEEL (fibrin glue) as a treatment for burn management. The study is composed of a histological analysis of rats skin after a 10 days burn model. The study is relevant to practice in burn management and research field. However, few details must be resolved before acceptance of the manuscript, specially in the description of the methods to ensure reproducibility and robustness of the data presented. Specific comments are as follows:
1) Title should indicate that this is a histological study in rats, since the study is very specific.
2) Introduction should be more descriptive on the rationale for using this fibrin sealant. Has it been used in wound healing before? What hypothetical or demonstrated benefits does it provide to support its use?
3) More description is needed in section 2.6. (histopathology). How many non-consecutive sections and fields were used per score / semiquantitative assessment?
4) H&E is not a good assessment to measure collagen deposition, since eosin would stain any extracellular matrix present. More specific staining like Picrosirius red or Masson’s Trichrome should be used to this end. Table 4 would refer as matrix deposition rather than specifically collagen
5) There is no description of the statistical analysis (despite that significant differences are indicated). Given that this is a semiquantitative assessment, I would guess that non-parametric analysis is required.
6) Table 5 represents the score of each rat. However, a single score is represented per N (no standard deviation nor mean), meaning that a single field was analyzed per specimen? Minimum of 5 sections (non consecutive) should be analyzed, and data should be expressed in median +/- quartile. Also, a representation in bar graph / box and whiskers is recommended.
7) Table 5, the significance found (p represented in the last column) is comparing only SYLFIO and TISSEEL? Or also the control? Or within the group using ANOVA? Specific differences between groups (post-hoc) should be included.
8) Discussion names a power analysis performed, this is not described in methods.
9) Fist paragraphs of the discussion ensure that fibrin glue has been shown to decrease inflammation, but in the present study, all the indicators show the opposite, contradicting the main point of the paper. Is this related to the time point chosen (10th day)?. If so, what is the rationale of choosing this time point?
Author Response
Please see the attached word file titled Response to Reviewer 2.

Reviewer 3 Report
Comments and Suggestions for Authors
The paper is interesting and properly elaborated. In my opinion it could be published in "Medical Sciences" after making some minor revisions suggested below:
In my opinion the statement that TISSEEL™ is a fibrin sealant should be included both in the title of the paper and in the text of "Abstract", in order to make the manuscript more informative for the reader who is not a specialist.
In section 2 "Materials and Methods" it should be explained why the animals were euthanized at 10th post burn day.
The mean value of weight of animals presented in "Abstract" (240+/- 20 g) differs from the mean value of animal's weight presented in section 3.1 Animals characteristics (236 +/- 10 g). It should be clarified. Moreover the commonly used abbreviation of unit gram is g instead of gr - it should be corrected.
The description of applied methods of statistical analysis is lacking - it should be added in section 2 "Materials and Methods".
Author Response
Please see the attached word file titled Response to Reviewer 3.

Round 2
Reviewer 1 Report
Comments and Suggestions for Authors
Accept
Reviewer 2 Report
Comments and Suggestions for Authors
Authors have addressed all the comments. The manuscript is ready for acceptance now.